# Nanoscale profiling of evolving intermolecular interactions in ageing FUS condensates
Alyssa Miller[1], Zenon Toprakcioglu[1], Seema Qamar[2], Peter St George-Hyslop[2,3,4], Francesco Simone Ruggeri [5] ✉, Tuomas P. J. Knowles [1,6] ✉ & Michele Vendruscolo [1] ✉

Protein condensates can exist in different states with distinct material properties, corresponding to specific cellular functions. These material properties, however, remain difficult to characterise, in part due to the technical challenges associated with studying condensed states. Here, to address this problem, we combine a microfluidic sample deposition technique that preserves the solution properties of condensates on surfaces with a nanometre-resolution spatial mapping method to characterise the time-dependent material properties of condensates of the fused in sarcoma (FUS) protein. This approach revealed two distinct phase transitions within FUS condensates. We first observed a spatially heterogeneous disorder-to-order transition initiating at the condensate interfaces and associated with intermolecular β-sheet formation. This process was then followed by the gelation of the condensate core, arising from an increase in the density of intermolecular interactions between intrinsically disordered regions. Overall, this study identifies molecular conformations associated with emergent phases of FUS condensates, and establishes a technology platform to understand the role of nanometre-scale phase changes in protein condensates.

Phase transitions of proteins have long been the subject of interest in molecular biology, from liquid to solid transitions associated with amyloid formation[1], to liquid crystal formation in crystallographic studies[2]. Recently, cells have been shown to exhibit protein phase transitions via the formation of biomolecular condensates[3,4].

A protein whose function has been linked to phase separation is the RNA-binding protein fused in sarcoma (FUS)[5–7]. The ability of FUS to form dynamic, fluid-like assemblies plays an important role in RNA metabolism and DNA damage repair processes[5–7]. This type of behaviour is imparted by certain sequence features, including a long, intrinsically disordered segment, which confers high levels of conformational freedom[8], and enrichment in particular amino acids, especially charged ones[9,10]. However, FUS exhibits further phase transitions characterised by the loss of fluid-like behaviour, which have been linked with amyotrophic lateral sclerosis (ALS)[11]. ALS-associated mutations have been shown to dysregulate these phase changes, resulting in the formation of large, insoluble assemblies, suggesting a possible pathogenic role of the liquid-to-solid transition[11].

More generally, as diverse states have been described for biomolecular condensates, it is becoming increasingly clear that material properties play an important role in their functions in health and disease[12–14]. Given these observations, it is desirable to develop diverse toolkits to characterise the material properties of biomolecular condensates. Efforts to understand the complex material properties of protein condensates are currently frustrated by the still limited number of quantitative techniques available for this purpose. The sensitivity to their environment, which is required for their biological function, is also what makes protein condensates particularly challenging targets, as it is difficult to prepare them in a state amenable to systematic studies[15]. Despite these problems, recent efforts employing structural biology, including nuclear magnetic resonance spectroscopy, scanning electron microscopy, optical microscopy and rheological approaches, revealed many details of the time-dependent changes in the material properties of biological condensates, with increasing evidence that phase transitions can occur heterogeneously within condensates[16–18]. However, there is still much to be learned about how these phases evolve within time and space.

[1]Centre for Misfolding Diseases, Yusuf Hamied Department of Chemistry, University of Cambridge, Cambridge, UK. [2]Department of Clinical Neurosciences, Cambridge Institute for Medical Research, University of Cambridge, Cambridge, UK. [3]Department of Medicine (Neurology) University Health Network and Tanz Centre for Research in Neurodegenerative Diseases, University of Toronto, Toronto, ON, Canada. [4]Department of Neurology (Carol and Gene Ludwig Center for Research on Neurodegeneration), Columbia University, New York, USA. [5]Physical Chemistry and Soft Matter, Wageningen University & Research, Wageningen, the Netherlands. [6]Cavendish Laboratory, Department of Physics, University of Cambridge, Cambridge, UK. ✉e-mail: simone.ruggeri@wur.nl; tpjk2@cam.ac.uk; mv245@cam.ac.uk

Here, we describe a strategy based on atomic force microscopy (AFM) that is capable of providing nanometre-resolution maps of the mechanical properties of individual condensates. We implement this approach through the use of a microfluidic sample deposition method that facilitates the application of surface-based techniques to study biological condensates[19–21] (Fig. 1). We first demonstrate that we preserve solution properties of condensates on surfaces. Then, using single-condensate AFM and complementary bulk infrared spectroscopy, we characterise the heterogeneous changes in condensate structure and mechanical properties over time to reveal local phase transitions in FUS condensates.

## Results

### Microfluidic spray deposition of protein condensates on surfaces

Protein condensates, including those formed by FUS, are notoriously difficult to deposit on surfaces without disrupting the complex interaction network that stabilises the condensed state[15]. We have previously developed a microfluidic spray deposition method that is capable of preserving the conformational state of biomolecules as in solution, including both globular and intrinsically disordered proteins[19,21]. Briefly, the method involves the precise generation of picolitre-volume droplets with known dimensions containing the biomolecules, which undergo ultra-fast drying (Fig. 2a). This ultra-fast drying is on the millisecond timescale, thus minimising the time for aberrant conformational changes to occur on the surface (Fig. S1)[19]. Therefore, we first sought to assess the hydration state of condensates following deposition. We considered the ratios of the peak intensity of the protein backbone (amide II; NH bending and CH stretching, ~1550 cm$^{-1}$) versus water (OH stretching, ~3400 cm$^{-1}$) measured via Fourier transform infrared spectroscopy (FTIR). Due to the removal of bulk water via evaporation, we assumed the origins of this water peak arise primarily from hydration within the condensate, with additional contributions from the condensate-air interface and the ever-present hydration layer between the IR prism and sample. We observed that freshly formed condensates (ageing time, $t_a = 0$ h) retain water within the dense phase, even after evaporation of bulk water, which is lost as condensates age (Fig. 2e). This finding is consistent with previous observations of high water content in liquid-like condensates, which appears to be less relevant in more solid-like assemblies[22].

We next explored the deposition of the condensates onto surfaces via microfluidic spray deposition, in order to preserve the relevant structural features, including morphology and heterogeneity (Fig. 2). To achieve this purpose, we first imaged condensates in solution via fluorescence microscopy, and compared the same solution deposited via a microfluidic spray device (Fig. S2). However, under the conditions chosen here, which were selected to be near physiological and are just above the saturation threshold in vitro (2 μM FUS, 75 mM KCl)[23], the resultant condensates were too small to be readily visualised by light microscopy. Therefore, we turned to high-resolution AFM to assess the morphological properties of the samples.

When using microfluidic spray deposition, we observed the presence of both the dense and dilute phase, indicating a preservation of the heterogeneity of complex mixtures. We also ensured that the protein was not retained in the microfluidic device by imaging the channels after spraying, where minimal material was observed (Fig. S3). Further, condensates deposited via the microfluidic spray procedure had round morphologies with evidence of fusion events, which are characteristic features of fluid-like condensates in solution. Therefore, we sought to quantify this effect by measuring the circularity of condensates (Fig. 2f). We next compared sample properties between microfluidic spray deposition and manual deposition. Briefly, manual deposition involves adding a volume of sample onto a surface, waiting to allow adsorption to the surface, followed by a rinsing step to remove excess material. When deposited manually, in our hands, we no longer observed the presence of the disperse phase, indicating that the rinsing step removes weakly adsorbed material. Remaining condensates displayed an amorphous, non-circular morphology, which is consistent with surface-induced aggregation of the dense fluid-like phase.

The preservation of relevant structural features can be rationalised by considering deposition times associated with each method. While manual deposition is typically performed on the second to minute timescale, spray deposition, due to the significant reduction in droplet volume, occurs on the tens of millisecond timescale. To contextualise these considerations, the relaxation time of liquid FUS condensates have been reported to be in the range of tens of milliseconds, which are comparable to the deposition timescales we employ here[24]. Therefore, there is less time for significant surface-induced structural rearrangement of network-spanning properties of the condensates, which is observed using manual deposition.

Moreover, it should be noted that since FUS condensates are stabilised by a complex combination of electrostatic and hydrophobic interactions, it was important to identify a suitable surface for imaging. Significant sample deformation was observed for charged surfaces, such as mica, which is typically used in AFM studies (Fig. S4). Finally, we determined that the use of hydrophobic zinc selenide crystals was suitable as we did not observe deformations of sample morphology, and it has minimal surface defects which can impede high-resolution imaging, in agreement with previous reports[10]. However, some alignment of the sample can be observed in surface crevices (Fig. 2c, d).

### Morphological characterisation of condensate ageing

Having established the ability to preserve relevant structural features of condensates on surfaces, we next sought to determine whether we could reliably measure time-dependent changes in condensate properties on a surface. Therefore, we asked how condensate morphology changes as a function of ageing. At the near-$c_{sat}$ conditions chosen here, significant heterogeneity is observed in solution, with the co-existence of small clusters and larger condensates[25,26]. This heterogeneity can be challenging to accurately capture via solution-based techniques, therefore making AFM well-

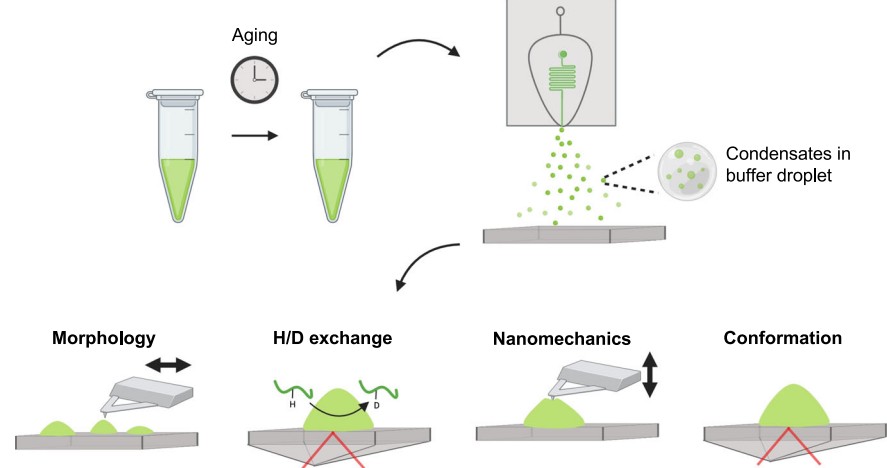

**Fig. 1 | Deposition of ageing FUS condensates via microfluidic spraying and characterisation of condensate material properties.** Overview of the experiments employed in this study. Condensates were deposited using a microfluidic spray device, which enables their characterisation via surface-based techniques. Four properties of condensates were probed as a function of ageing: (1) 3D morphology via AFM, (2) hydrogen-deuterium exchange rates via IR spectroscopy, (3) material properties via AFM nanomechanical mapping, and (4) chemical and structural properties via IR spectroscopy. This combined approach allowed us to generate a detailed model of how phases emerge in space and time within FUS condensates, and how these phases correlate with conformational changes at the polypeptide level.

Aging

Condensates in buffer droplet

Morphology    H/D exchange    Nanomechanics    Conformation

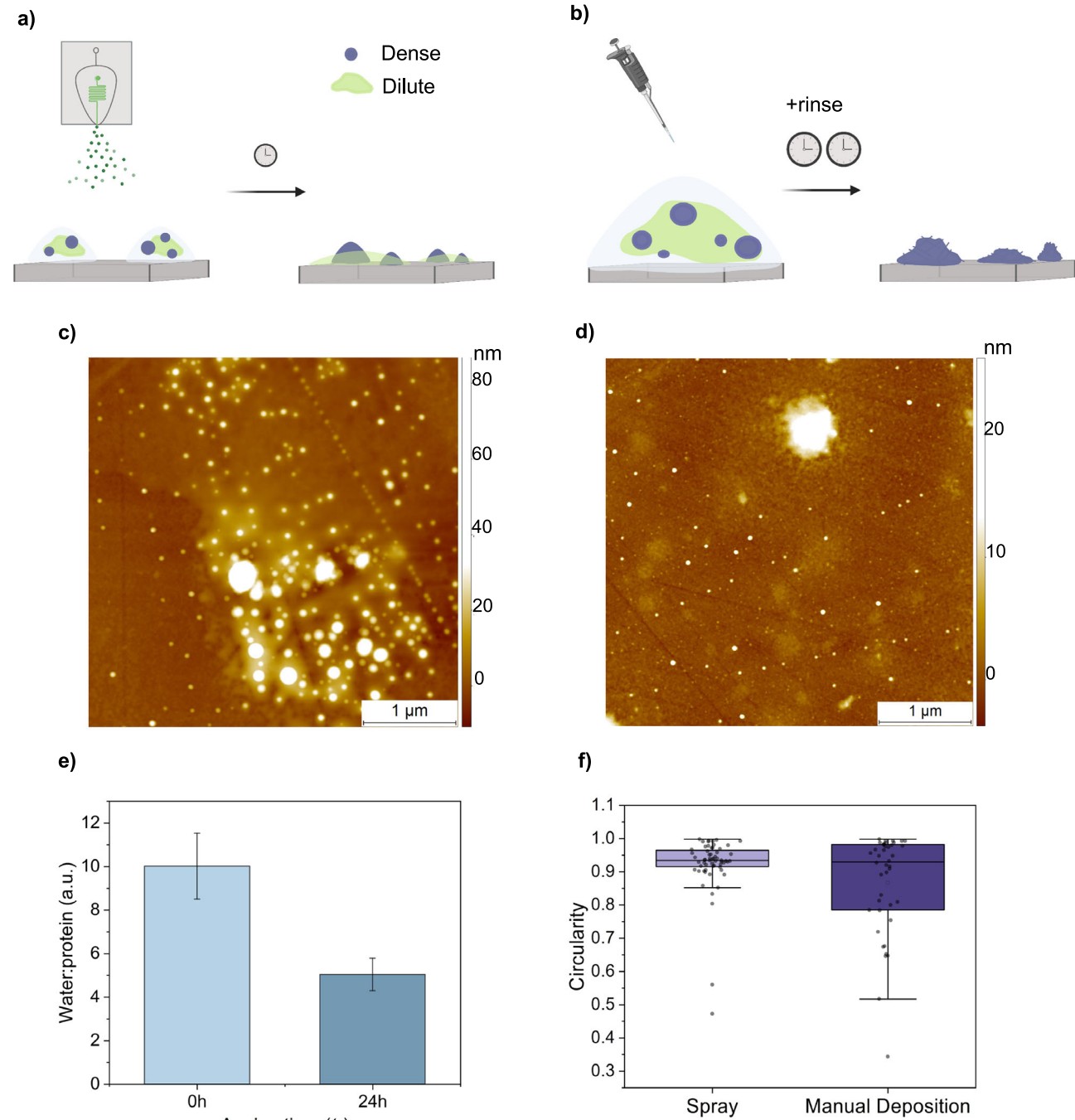

**Fig. 2 | Comparison of surface deposition methods for the characterisation of protein condensates.** Schematics of the sample preparation method for manual deposition (**a**) and microfluidic spray deposition (**b**). Manual deposition requires long incubation times (seconds to minutes) and a rinsing step. Spray deposition can be carried out in a single step with short incubation times (tens of milliseconds). AFM images were acquired for manual (**c**) and spray **d** deposition, where morphological differences can be observed. **e** The hydration state of freshly formed and aged condensates was measured by considering the ratio of water to protein, measured via integration of IR peaks at ~3400 cm$^{-1}$ (water, O–H bending) and ~1550 cm$^{-1}$ (protein, N–H bending and C–N stretching). These ratios do not reflect stoichiometry in the condensates, as IR peak intensities vary as a function of wavenumber ($n = 3$ per condition, technical replicates). Error bars represent SD. **f** Circularity was compared for condensates deposited via microfluidic spray deposition and manual deposition ($n = 60$ and 42 condensates for spray and manual deposition, respectively; two sample preparations per condition). Box represents 25, 75 percentile and whiskers represent SD.

suited as it can resolve single assemblies, from the nanometre to micrometre scale. To this end, FUS samples were incubated, and aliquots were taken at distinct time points (ageing time ($t_a$) = 0, 2, 4, 8, and 24 h) and deposited for surface characterisation (Fig. 1). We first imaged sample morphology via confocal microscopy, which has the benefit of a large field of view, and thus provides an initial overview of the sample properties (Fig. S2). At $t_a = 0$, 2 and 4 h, very few, dim condensates are observed. At $t_a = 8$ and 24 h, more

large condensates are observed, which have a higher fluorescence intensity. To characterise this feature in depth beyond the diffraction limit of light, we also measured condensate morphology via AFM (Fig. 3).

At the initial time point, $t_a = 0$ h, we indeed observed large condensates as well as numerous small, spherical species, which are consistent with previously described clusters, as expected[25,27]. These clusters did not persist beyond $t_a = 0$ h, and condensate size was observed to increase up until

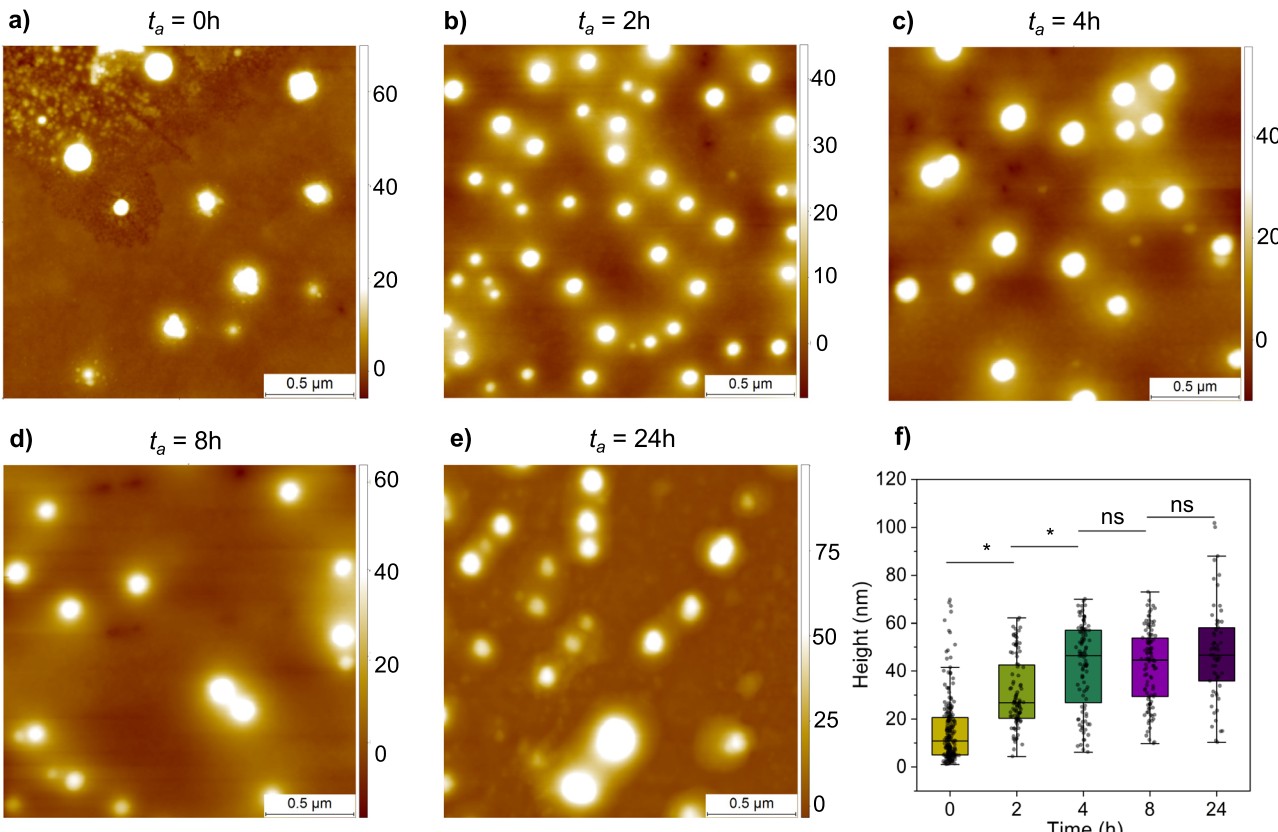

**Fig. 3 | Characterisation of age-related changes in the condensate morphology. a–e** The morphology of condensates was measured via AFM as a function of the ageing time ($t_a$). Z-scale bar is in nanometres. **f** Single condensate statistical analysis was performed on AFM images to determine change in height as a function of ageing time. Mean heights and SD: 0 h 15 ± 14 nm, $n$ = 224 condensates from 3 sample preparations; 2 h, 31 ± 15 nm, $n$ = 101; 4 h, 42 ± 18 nm, $n$ = 104; 8 h, 42 ± 15 nm, $n$ = 105; 24 h, 48 ± 20 nm, $n$ = 62. ns not significant, *$p$ < 0.05, one-sided $t$ test. Box represents 25, 75 percentile and whiskers represent SD.

$t_a$ = 4 h, after which no significant size increase was further detected. There are various theories as to why condensate growth via fusion may halt, including internal structural changes associated with ageing (i.e disorder to order transitions)[28], or changes in rigidity of the condensates scaffold[29]. Therefore, we sought to explore this via nanoscale-resolution nanomechanical mapping via AFM.

We note here that amyloid fibrils were reported to form from the low complexity domain of FUS[30–32] however, we did not observe instances of this type of structure in the present work with full-length FUS (Fig. S5). Indeed, as FUS solid assemblies were previously described to reach sizes incompatible with the size of the microfluidic device channels (Fig. S3), we are only able to report on the properties of assemblies below 20 μm. Therefore, we also manually prepared samples at $t_a$ = 24 h via pipetting. In these samples, we observed the presence of micrometre-scale fibrous structures (Fig. S6). However, it is difficult to disentangle the role of shear forces from pipetting on fibre formation, especially as fibres are rarely reported to form from full-length FUS in the time-scales employed here[33,34]. Therefore, we do not focus on the characterisation of these fibrous structures in this work, and rather report on the intermediate structures which precede fibre formation, which are less well characterised.

**Sub-condensate mapping of time-dependent material properties**
To further understand the changes in the material properties of FUS condensates with ageing, we directly measured their mechanical properties in an aqueous environment. The Young's modulus, or elastic response, of condensates can be measured by considering how they respond to local deformations by the AFM tip, via the acquisition of force-distance curves (Fig. 4a, b)[35]. Since the elastic response to deformation is a characteristic behaviour of solid materials, the Young's modulus provides a specific readout on the emergence of solid-like behaviour, which is characteristic of a disorder-to-order transition.

Therefore, we report on the elastic modulus as a function of ageing time from $t_a$ = 2 h, based on the acquisition of force-distance curves taken in the centre of condensates (Fig. 4c). These measurements reveal that at early time points, condensate cores have elastic modulus values of 9 ± 5 kPa and 21 ± 9 kPa for $t_a$ = 2 h and 4 h, respectively. At $t_a$ = 8 h, the elastic modulus is more variable, with values of 37 ± 31 kPa. At $t_a$ = 24 h, values can be estimated to be about three orders of magnitude higher (1.1 ± 2.2 MPa) than those at $t_a$ = 2 h. We note that the stiffness of our probe (0.7 N/m) did not allow us to stably measure the elastic response of fluid condensates below the KPa range, meaning that we may not readily report here on the mechanical properties of condensates at $t_a$ = 0 h. Additionally, the probe is too soft to accurately measure mechanical properties of the substrate, which has stiffness in the tens of GPa range, thus, the background was removed.

The Young's modulus values aquired here via AFM are generally within the same range as previously described by solution-based measurements, albeit a bit higher, which may be explained by the large inherent error of ~8% due to approximations in the contact model employed (Supplementary Note 1), as well as the relatively high tip loading rate (and therefore the Young's modulus here likely also reflects some viscoelastic contributions)[16]. Nevertheless, at a constant tip loading rate, our approach is appropriate to study relative changes in Young's modulus, such as the time-dependent changes we study here. The increase in Young's modulus we detect with ageing reveals an emergent elastic component of FUS condensates. However, it is difficult to ascertain whether these changes are arising solely from a disorder-to-order transition, yielding solid structures, as FUS condensates display complex age-related material properties, with both elastic and viscous behaviour. Thus, these Young's moduli values likely

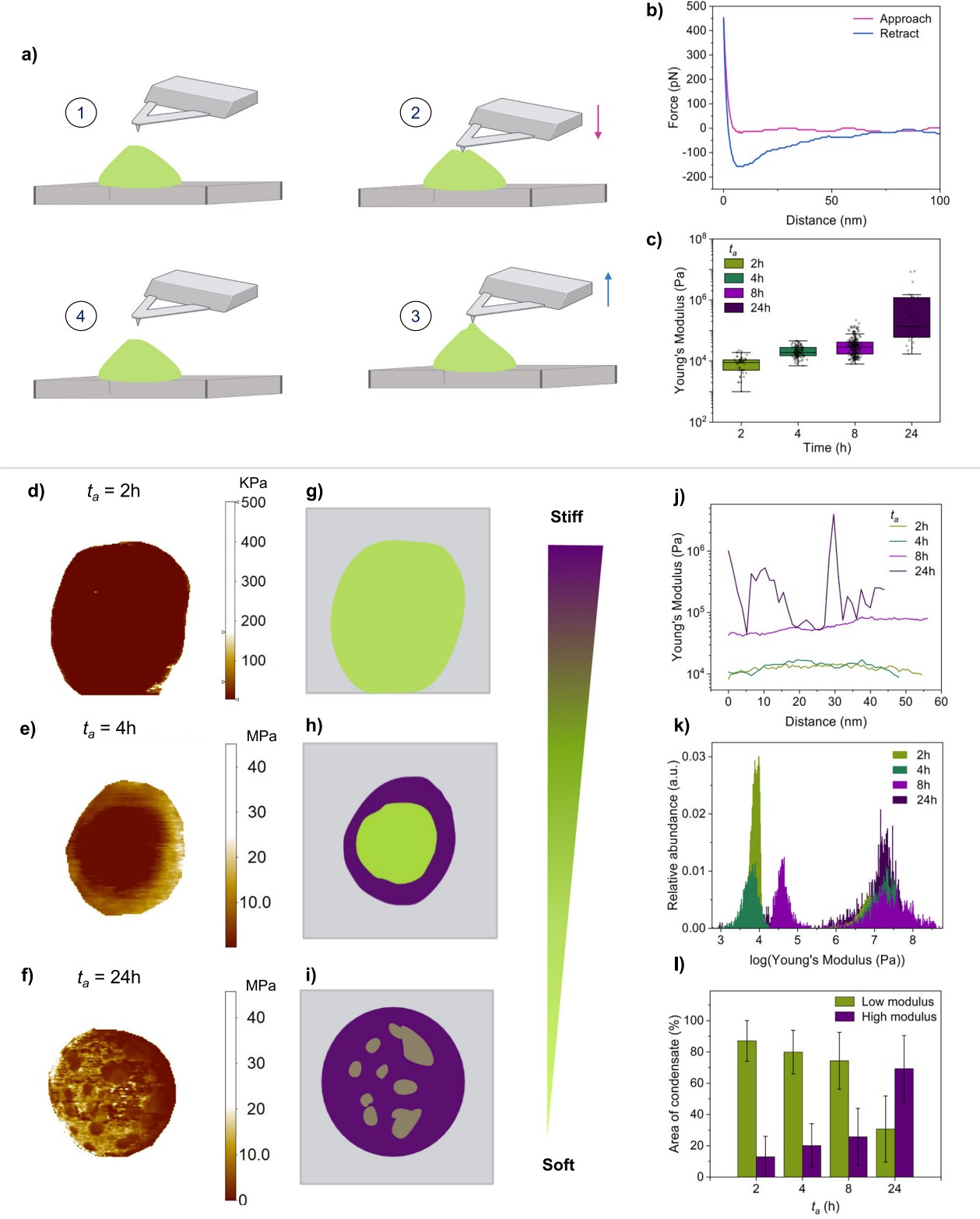

also reflect contributions from changes in the viscoelastic behaviour, such as those described to arise from gelation[36].

To further understand the network-level changes giving rise to the increase in Young's moduli, we next performed spatially resolved nanomechanical mapping to complement the force-distance measurements acquired in the condensate cores[37,38]. We reasoned this would enable us to discriminate between gelation and solidification, as solid-structures have

been observed to emerge heterogeneously within condensates, which we can therefore detect via spatially resolved nanomechanical maps[16,18] (Figs. 4d–f and S7)[39]. As this method requires a physical deformation of the condensates, we first established that measurements were not destructive. To this end, we performed repeat measurements of single condensates to assess for changes in morphology and material properties, both of which serve as an indicator of deformation-induced aggregation. No significant

**Fig. 4 | AFM mapping of the material properties of single condensates.**
**a** Schematic showing the procedure of nanomechanical characterisation using an AFM, to generate a force-distance curve. **b** A representative force-distance curve is shown for $t_a$ = 24 h. **c** Young's modulus values of single condensates shown as a function of time. Each point corresponds to the average of three force-distance curves, taken in the centre of condensates (2 h, $n$ = 51; 4 h, $n$ = 117; 8 h, $n$ = 243; 24 h, $n$ = 27, from 3 independent sample preparations). Box represents 25 and 75 percentiles, error bars represent SD. Representative AFM nanomechanical maps of the Young's modulus in log scale are presented for individual condensates as a function of ageing: $t_a$ = 2 h (**d**), 4 h (**e**) and 24 h (**f**). Corresponding non-quantitative schematics are shown for each nanomechanical map **g–i** to serve as a visual aid. Green represents regions with lower elastic modulus (soft) and purple represents regions with higher elastic modulus (stiff). **j** Representative cross-sections of the Young's modulus in the core of condensates are plotted. This reveals a relatively uniform distribution of material properties within the condensate centre up until $t_a$ = 24 h, in which significant heterogeneity emerges in some condensates. **k** Histogram of the Young's modulus values within a single condensate are shown as a function of ageing. A height threshold was applied to exclude the surface, meaning only the area of the condensate is considered. **l** The emergence of the high elastic modulus was quantified as a function of time, by measuring the area of the high modulus phase over the total area of the condensate. The low modulus phase is <1 MPa, and the high modulus phase is >1 MPa. Error bars represent SD.

changes in either property were observed, enabling us to proceed with time-resolved nanomechanical mapping of condensates (Fig. S8). At $t_a$ = 2 h, we observed condensates to be materially uniform, with a low elastic modulus in the same range as that measured via force-distance curves in the condensate cores. At $t_a$ = 4 h and 8 h, we began to observe the emergence of a high elastic modulus shell at the edge of the condensates. Importantly, AFM nanomechanical measurements display an inherent artefact related to substrate effects, in which mechanical properties of a sample are not accurately measured at its thinner boundaries[40]. This is particularly relevant in instances of soft samples on hard supports, as used here, as the measured Young's moduli will include a combination of contributions from the soft sample and the hard substrate below, due to the round shape of condensates leading to lower heights at their edges. Thus, to decouple surface effects from the phenomena of interest, we measured the thickness of the high-modulus shell as a function of time. One would expect the condensates at $t_a$ = 2 h to display the largest edge effects, due to their low elastic moduli. However, in fact, we observed the opposite, with a thicker shell at $t_a$ = 4 and 8 h (Fig. S9). Even while considering edge effects, this high-modulus phase emerging at condensates edges is consistent with heterogeneous solid formation associated with disorder-to-order transitions, as previously reported[16,17]. Finally, at $t_a$ = 24 h, we observed co-existing low and high-modulus phases throughout the condensate core in some condensates, and some condensates which were uniformly high modulus (Figs. 4f, j and S10).

From spatially resolved nanomechanical maps, we sought to quantify the distribution of phases within single condensates. We therefore plotted histograms of the Young's moduli of whole, single condensates to enable quantification of both the high modulus phase and the lower modulus phase. (Figs. 4k, l and S10) At $t_a$ = 2 h, there is one dominant peak corresponding with a low elastic modulus phase. At $t_a$ = 4 h, there are two dominant peaks, indicating the presence of a dominant, low elastic modulus phase in the condensate core, and a less abundant, high elastic modulus phase corresponding to the solid formation at the condensate-solvent interface. At $t_a$ = 8 h, this same high elastic modulus (solid) phase exists. However, the elastic response of the soft core increases. Finally, at $t_a$ = 24 h, the dominant peak is the high elastic modulus (solid) phase. These results indicate that two distinct ageing processes are occurring within single FUS condensates: a heterogeneous disorder-to-order transition initiated at the condensate edge, and a uniform change in the elastic response of the condensate core, which is typical of gelation. Thus, we next aimed to understand the polypeptide-level conformational changes underlying both of these phase transitions.

## Age-dependent changes in conformation and secondary structure

Having characterised two parallel ageing processes occurring in ageing FUS condensates, we next sought to determine whether there were corresponding changes in secondary structure and conformation via FTIR. The use of this technique is greatly facilitated by the microfluidic spray deposition. Ultra-fast deposition times minimise salt crystallisation, thereby reducing the contribution of buffer salts to the spectra[19]. This affords a significant increase in spectroscopic sensitivity, such that we can extract relevant features which would typically be difficult to distinguish from noise, and measure secondary structure in physiologically relevant high salt conditions (75 mM KCl). This is important, as FUS phase behaviour is intrinsically linked to the protein concentration and ionic strength[41].

At $t_a$ = 0 and 2 h, where the dominant phase is the uniform low-modulus component, the IR spectrum was dominated by α-helical structure, which has a characteristic sharp peak at 1657 cm$^{-1}$, and random coil structure at 1648 cm$^{-1}$ (Fig. 5b, c, e,) assignment of which is assisted by comparing second derivative spectra in H$_2$O and D$_2$O (Fig. S11). This is further confirmed by a downward shift of the amide I band of ~10 cm$^{-1}$ in D$_2$O, which is characteristic of random coil structure (Fig. 6a). A similar spectrum was observed at $t_a$ = 4 h, albeit with the observation of a subtle shoulder related to intermolecular β-sheet peak at 1629 cm$^{-1}$. These structural features are in line with solution-structures of FUS, which reveal a long, intrinsically disordered domain as well as an α-helical, RNA-recognition domain[42]. This is also consistent with reports that the transition from the dilute to dense phase is associated with minor changes in secondary structure[22]. Additionally, the width of the amide I peak can be used to understand conformational flexibility. As each spectrum represents the averaging of snapshots of the conformations of each polypeptide in bulk, a broad peak indicates the averaging of many conformations. (Fig. 5d)[8]. Notably, there is a native β-sheet peak at 1638 cm$^{-1}$, attributed to the β-barrel of GFP tag[43]. A tagged protein was used to enhance protein solubility for ease of handling and to facilitate comparison with light microscopy experiments.

As $t_a$ increases, corresponding with the formation of a spatially heterogeneous solid phase, we observed a new clear peak at 1625 cm$^{-1}$ indicating the presence of intermolecular β-sheet (Fig. 5c). The intensity of this peak increases between $t_a$ = 8 and 24 h. We can further understand the nature of the intermolecular β-sheet assemblies here by considering both the peak position and width, which provides information on chemical environment and conformational heterogeneity. Peaks of intermolecular β-sheets are typically in the range of 1625–1610 cm$^{-1}$, with a higher density of intermolecular hydrogen-bonding resulting in a shift to lower wavenumbers[44]. Furthermore, the whole amide I peak width only decreased by ~1.5%, compared to spectra from $t_a$ = 0, 2 and 4 h, which is within the error of the measurements. Taken together, this is consistent with a high degree of conformational heterogeneity being retained, even in the solid-phase.

Having established the secondary structure associated with the low modulus and solid phases, we next sought to assess the interaction network involved. FTIR spectroscopy is also capable of measuring the chemical environment of side chains, and therefore can be used to infer their interactions. Here, we focussed on tyrosine residues, which are enriched in the LCD, and are believed to be key drivers of phase separation due to their involvement in cation-π interactions with arginine residues[9,10]. There are 36 tyrosine residues in FUS: 24 in the LCD, 2 in the folded RNA-recognition motif, 9 in the arginine-glycine-glycine rich (RGG) motif stretch, and 1 in the nuclear localisation signal NLS. Tyrosine can be monitored via the C=C stretching vibration of the aromatic ring, which has a characteristic peak at ~1517 cm$^{-1}$ (Fig. 5e)[45]. To increase the accuracy of the peak assignment, we also compared spectra to those acquired in D$_2$O, where we see the expected downward shift to ~1514 cm$^{-1}$ (Fig. S12)[45]. At later time points of ageing, corresponding with the formation of solid-like, β-sheet assemblies, we

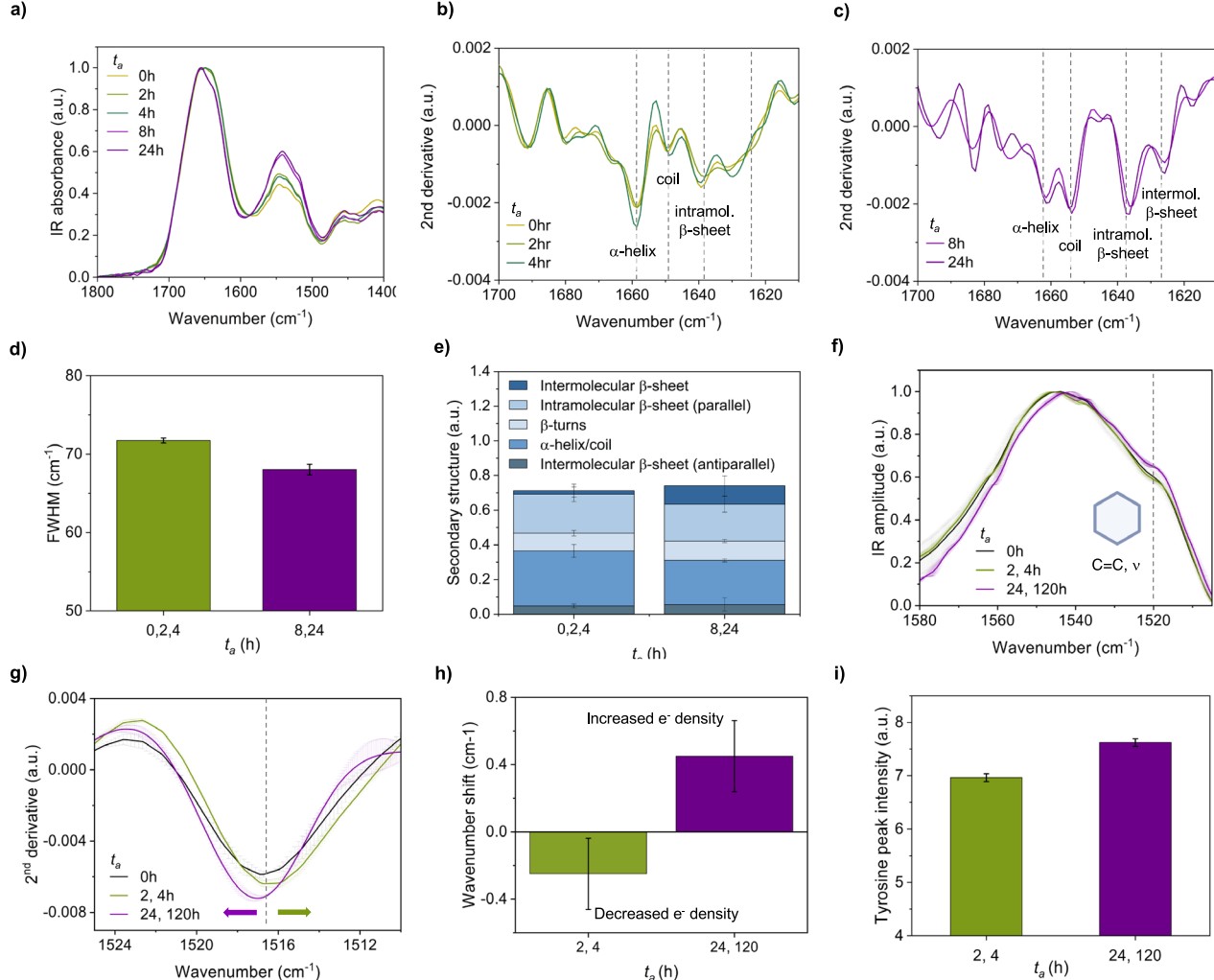

**Fig. 5 | Secondary structure changes associated with FUS ageing. a** The secondary structure of FUS condensates was measured as a function of time using FTIR, at $t_a = 0$ to 24 h. The second derivative of the amide I bands are presented for early, $t_a = 0$, 2 and 4 h (**b**) and late, $t_a = 8$ and 24 h **c** condensates. Differences in secondary structure components are highlighted based on characteristic absorption peaks. **d** The width of the amide I peak was measured, FWHM = full width at half maximum. **e** The relative amounts of secondary structure were quantified for early, $t_a = 0$, 2 and 4 h (**c**) and late, $t_a = 8$ and 24 h, based on the second derivative spectra. **f** Tyrosine residues, and their chemical environment, can be measured via FTIR. The tyrosine absorption peak for the liquid/dense phase is at ~1517 cm$^{-1}$, indicated by the black dashed line. Tyrosine peaks were also recorded for $t_a = 2$ and 4 h (green) and solid $t_a = 24$ h and 5 d (purple) samples. A different chemical environment will result in shifts of the peak. **g** The second derivative of the tyrosine peak reveals downward and upward shifts for early-aged condensates ($t_a = 2$ and 4 h, green) and late-aged condensates ($t_a = 24$ and 120 h, purple), respectively, relative to the peak position of at $t_a = 0$ h (black). The black dashed line indicates the peak position at $t_a = 0$ h. **h** The shifts in the tyrosine peak relative to $t_a = 0$ h were quantified. **i** The intensity of the tyrosine peak was measured by considering the integral of the peak. Differences in the peak intensity at $t_a = 24$ and 120 h indicates a different chemical environment. All measurements were taken from 3 technical replicates. Error bars represent SD throughout.

observed an upward shift of the tyrosine C=C peak relative to that at $t_a = 0$ h, consistent with an increase in the electron density in the aromatic ring (Fig. 5f, g)[45]. As one possible scenario, we suggest that this may be due to hydrogen-bonding taking place between aromatic groups in the solid phase, and is consistent with the traditional understanding of the role of intermolecular hydrogen bonding in β-sheet secondary structures[44]. We also observed an increase in the intensity of the tyrosine peak at later time points of ageing (Fig. 5h), again suggesting a different chemical environment of these residues. Interestingly, we observed a distinct tyrosine chemical environment at the earlier time points of ageing associated with the low-modulus phase. This is evidenced by a downward shift at $t_a = 2$ h, relative to 0 h, i.e a decrease in the electron density in the aromatic ring, opposite to the trends in the solid phase. This is consistent with an increase in the electron donation from the tyrosine to an electron-accepting group, which is likely due to an increase in the density of cation-π interactions, as has been previously reported, although this warrants further investigation[46]. The electron density further decreases at $t_a = 4$ h, which is consistent with a time-dependent increase in the density of the intermolecular bond network. This is consistent with the progressive increase in the Young's modulus of the condensate core as measured via AFM. At $t_a = 8$ h, in which both dynamic and solid phases are observed (Figs. 4 and S10), the peak position is approximately the same as at $t_a = 0$ h, with only a slight downward shift, likely indicating an averaging of the two relevant interactions in the co-existing low-modulus and solid phases observed at the condensate core and interface, repsectively (Fig. S13). Overall, based on the distinct chemical environments experienced by tyrosine residues, we suggest a different form of interaction appears to be relevant in stabilising the fluid and solid phases.

To further explore which regions of the FUS sequence are involved in these evolving interaction networks, we sought to overcome the inability of bulk FTIR to resolve specific residues. Thus, we turned to hydrogen-deuterium exchange (HDX) via FTIR, which provides information on the chemical environment, and therefore bonding networks, taking place in the

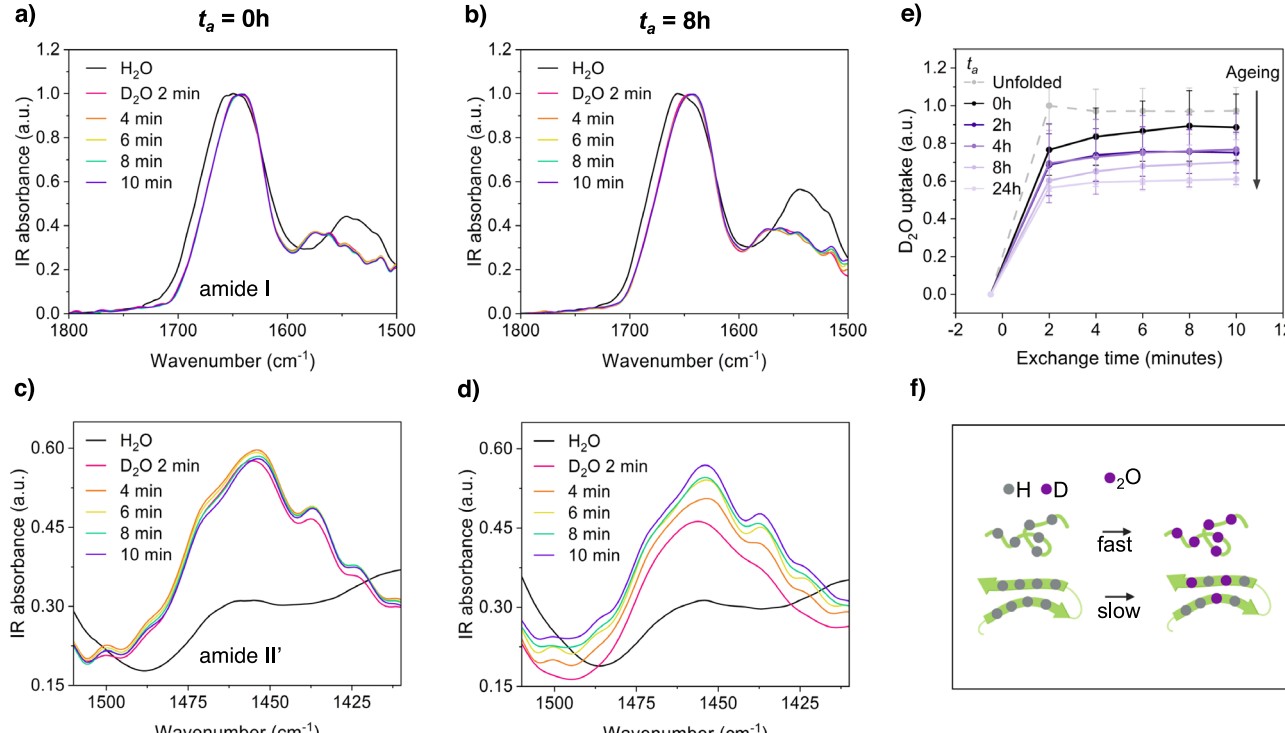

**Fig. 6 | Hydrogen-deuterium exchange provides information on the conformational dynamics within condensates.** Bulk hydrogen lability of protein condensates as a function of $t_a$ was investigated using hydrogen-deuterium exchange via FTIR. Condensates were deposited on the FTIR prism at different $t_a$, and rate of exchange was measured. HDX can be monitored by changes in the amide I (~1650 cm$^{-1}$), amide II (~1550 cm$^{-1}$) (**a, b**) and amide II' (~1450 cm$^{-1}$) (**c, d**). Black lines represent spectra in H$_2$O, and coloured lines show changes in spectra after transfer into D$_2$O. Representative spectra are presented for $t_a$ = 0 h (**a, c**) and 8 h (**b, d**); the spectra for the other time points can be found in the supplementary information. **e** The extent of exchange was monitored over the course of 10 min by taking the integral of the amide II' peak to represent D$_2$O uptake. Each sample was internally normalised against the intensity of the amide II peak in H$_2$O to control for baseline intensity differences, and against denatured FUS. Error bars represent SD **f** schematic displaying the conformation-dependence on backbone proton-lability: protons in random coil regions exchange rapidly, while those involved in hydrogen bonding and/or folded domains are less labile [44]. All measurements taken from 3 technical replicates.

polypeptide backbone (Fig. 6). Backbone protons engaged in folded secondary structure (such as α-helix, or β-sheet to a greater extent) exchange more slowly than unfolded regions due to the influence of hydron-bonding. Therefore, we used HDX to map which region of the sequence was involved in the interaction networks in the low modulus and solid phase, based on their solvent accessiblity. Briefly, exchange from a hydrogenated state to deuterated state can be monitored via 3 changes in the protein amide peaks: i) the amide I (~1650 cm$^{-1}$) peak will experience a shift to lower wavenumbers, which is especially prominent for random coil structure, ii) the amide II (~1550 cm$^{-1}$) peak will diminish and be replaced by iii) the emergent amide II' (~1450 cm$^{-1}$) peak, which increases in intensity with increased deuteration[47]. We thus monitored the change in the amide II' peak over time in D$_2$O, as a function of condensate ageing ($t_a$).

We chose short exchange times to preserve protein stability[48,49], and to minimise the influence from the folded domains, where exchange occurs on longer timescales. Further, we denatured the protein and saw only a slight increase in the D$_2$O uptake, therefore, we assumed that the exchange signal arises primarily from the intrisincally disordered regions of the protein, i.e the LCD and RGG motifs. We observed a progressive decrease in D$_2$O uptake as a function of $t_a$ (Fig. 6e), as well as a variable decrease in the rate of exchange (Figs. 6c, d and S14). This decrease in backbone hydrogen-lability of aged condensates is consistent with the intrinsically disordered regions of FUS being involved in progressively stronger interaction networks.

## Discussion

In this work, we presented a characterisation of protein condensates on surfaces through a combination of AFM-based nanomechanical mapping and bulk FTIR using a microfluidic spray deposition method. We showed that it is possible to preserve relevant conformational features of FUS

condensates on surfaces, thus enabling us to perform high-resolution structural and mechanical characterisation of the ageing behaviour of these condensates.

The enhanced capability of studying condensates on surfaces generated a detailed model of how solid-like features temporally and spatially emerge within initially liquid-like condensates, and determine how these phase changes correlate with conformational changes at the molecular level (Fig. 7). Initially, we observed the presence of uniform condensates with fluid-like properties imparted by the intrinsically disordered conformation of component protein molecules. An increasingly elastic, solid-like, intermolecular β-sheet-rich phase then formed at the condensate-solvent interface, while the condensate core remains fluid-like, which is consistent with previous reports[16,17]. In parallel, there was also an increase in the density of intermolecular interactions between intrinsically disordered regions of FUS molecules within the fluid core, resulting in an increase in the viscoelastic response. Finally, the solid phase heterogeneously permeated from the condensate edges through the aged fluid core. Spectral analysis suggested conformational heterogeneity and low-density intermolecular interactions between cross-β sheets within the solid phase. We also observed that tyrosine residues adopt a different conformation in these solid assemblies, indicating that they are involved in different interaction networks. Tyrosine residues were shown to be required for transcriptional activity, and thus their different molecular environment in solid assemblies may affect their accessibility, perhaps in connection with the pathological nature of aged FUS assemblies[11,51]. However, much of the information on interactions networks was derived from bulk FTIR, supported by Young's modulus measurements via AFM. Therefore, further exploring our findings through high-resolution, single-molecule measurements would be beneficial.

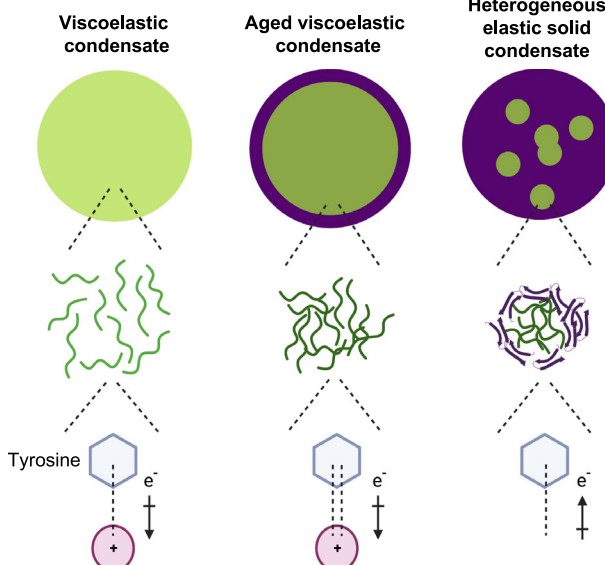

**Fig. 7 | Schematic representation showing the local phase transitions in ageing FUS condensates.** FUS condensates can be initially characterised as viscoelastic materials, as they exhibit both viscous (i.e liquid-like) and elastic (i.e solid-like) characteristics when undergoing deformation[50]. Upon ageing, two phase transitions are observed: (1) a gelation transition associated with increase density of inter-molecular interactions between intrinsically disordered molecules at the condensate core, and (2) a dominant disorder-to-order transition associated with β-sheet structure formation, giving rise to solid-like behaviour, which is promoted at condensate-solvent interfaces, and eventually propogates heterogeneously through the condensate.

Biomolecular condensates have complex material properties, thus necessitating the development of diverse methodologies to capture the full breadth of these complex material states[52–56]. This synergy between methods led to the emergence of a picture of the heterogeneous process of FUS condensate ageing. Our findings complement those from more established technologies, thus expanding the toolkit to study biomolecular condensates. By performing structural studies using surface-based techniques, we are no longer limited by polypeptide length, as some solution-based methods are, allowing us to study full-length FUS. This experimental procedure allowed us to build on the valuable insight provided from structural studies on particular domains, such as the low complexity domain, giving us an understanding of how the overall properties of the full protein sequence may alter the ageing behaviour of FUS condensates[31]. We also anticipate these methods may be particularly useful for proteins whose phase behaviour is broadly affected by fluorescent tagging, as it does not rely on optical detection.

## Materials and methods
### Production and purification of full-length FUS protein
FUS protein was purified as previously described[10]. Constructs encoding FUS residues 1-526 were cloned into pACEBac2 vector with an EmGFP C-terminal tag. Proteins were expressed and purified from insect Sf9 cells infected with the baculovirus. After four days of infection, cells were harvested by spinning at 4000 rpm for 30 min. Cell pellets were mixed with the resuspension buffer containing 50 mM Tris, 1 M KCl, 0.1% CHAPS, 1 mM DTT, 5% glycerol at pH 7.4, and proteins purified using three steps purification scheme including, Ni-NTA affinity column, Amylose affinity column followed by size exclusion chromatography in the buffer containing 50 mM Tris, 1 M KCl, 1 mM DTT, 5% glycerol at pH 7.4.

### Fabrication of microfluidic devices
A two-step photolithographic process was used to fabricate the master used for casting microfluidic spray devices[57]. In brief, a 25 µm thick structure

(3025, MicroChem) was spin-coated onto a silicon wafer. This was then soft-baked for 15 min at 95 °C. An appropriate mask was placed onto the wafer, exposed under ultraviolet light to induce polymerisation, and then post-baked at 95 °C. A second 50 µm thick layer (SU-8 3050, MicroChem) was then spin-coated onto the wafer and soft-baked for 15 min at 95 °C. A second mask was then aligned with respect to the structures formed from the first mask, and the same procedure was followed, i.e exposure to UV light and post-baking for 15 min at 95 °C. Finally, the master was developed in propylene glycol methyl ether acetate (Sigma-Aldrich) to remove any photoresist which had not cross-linked. A 1:10 ratio of PDMS curing agent to elastomer (SYLGARD 184, Dow Corning, Midland, MI) was used to fabricate microfluidic devices. The mixture was cured for 3 h at 65 °C. The hardened PDMS was cut and peeled off the master. The two complementary PDMS chips are then activated with $O_2$ plasma (Diener Electronic, Ebhausen, Germany) and put in contact with each other and aligned precisely such that the gas inlet intersects with the liquid inlet to form a 3D nozzle[19–21].

### FUS sample handling
Aliquots of FUS solution was thawed immediately before use. Phase separation was induced in an Eppendorf tube by lowering the salt concentration, via sample dilution to a final concentration of 75 mM KCl, 25 mM Tris. Aliquots were then taken from the solution at desired time points (i.e 0, 2, 4, 8 and 24 h) and promptly deposited via microfluidic spray deposition (described below). For measurements with manual sample deposition, 10 µl of sample was deposited on the surface and allowed to adsorb for 5 min. The sample was then rinsed with MilliQ water and dried with a gentle stream of nitrogen gas. For all measurements on the 5-day aged sample, manual deposition was performed, due to the viscosity of the sample preventing it from passing through the device channels.

### Microfluidic spray deposition
Prior to introduction of sample, each device was tested and washed with MilliQ water for 5 min. Sample was then loaded into 200 µL air-tight glass syringes (Hamilton) and driven into the spray device using a syringe pump (Harvard apparatus). Solutions containing sample were pumped into the device with a maximum flow rate of 100 µl/h to minimise sample shearing, while the nitrogen gas inlet pressure was maintained at 3 bar. Deposition was conducted for a maximum of 10 s at a distance of 3.5 cm to ensure coalescence of droplets did not occur. Samples were sprayed directly onto the relevant surfaces (i.e ZnSe crystals, FTIR prism) with no further washing steps required before measurements.

### Confocal microscopy
FUS condensates were deposited via microfluidic spray deposition, or via manual pipetting of ~20 µl onto a glass coverslip. Images were taken using a confocal microscope (Leica, TCS SP8). Excitation/Emission wavelengths of 405 nm/510 nm were used, respectively, to image the condensates. All images were taken under ambient conditions, and acquisition was initiated immediately after deposition, in order to minimise any evaporation, in the case of manual deposition.

### Atomic force microscopy
Samples were deposited onto ZnSe crystals via microfluidic spray deposition for morphology imaging. High-resolution morphology measurements were performed in ambient conditions on an NX10 AFM operating in non-contact mode (Park Systems, South Korea). PPP-NHCR probes were used, with a spring constant of 42 N/m and with a nominal tip radius of ~10 nm. Images were acquired at a scan rate of 0.3–0.5 Hz to minimise forces applied to the delicate condensates. Nanomechanical characterisation was performed in 25 mM Tris, pH 7.4 buffer using a MultiMode 8 (Bruker, USA) operating in either force-volume or quantitative nanomechanical mapping mode. ScanAsyst Fluid probes were used, with a spring constant of 0.7 N/m and a nominal radius of ~20 nm. Images were acquired at scan rates of 0.3–0.5 Hz, at 512 × 512 pixels. Probes were calibrated using the thermal

tune method. Peak force values were chosen between 200 and 1000 pN. Deformation was monitored to ensure consistent indentation values across condensates with different material properties, and to ensure indentation depths did not result in excessive sampling of the substrate (indentation depths were maintained below the 10% threshold of sample height). Only condensates above 30 nm in height were considered. All measurements were performed at room temperature. Scanning probe image processor (version 6.7.3, Image Metrology, Denmark) software was used for image flattening and single condensate statistical analysis. Background removal of nanomechanical maps was performed via masks which selected condensates based on heights from morphology maps. Force-distance curves were analysed in Nanoscope Analysis software (Bruker, USA). A Hertz-DMT contact mechanic model was used to fit the contact region of withdrawal curves. The contact region of the withdrawal curve was analysed via second derivative analysis; those curves which showed a biphasic response were rejected, as this was taken as an indication of contribution from the hard substrate.

## FTIR and HDX

Measurements were performed on a Vertex 70 FTIR spectrometer (Bruker, USA) equipped with a DiamondATR unit and a deuterated lanthanum a-alanine-doped triglycine sulfate detector. Each spectrum was acquired with a scanner velocity of 20 kHz over 4000–400 $cm^{-1}$ as an average of 256 scans. Thin films were deposited on the prism using microfluidic spray deposition, and the film was rehydrated via flushing with $H_2O$ vapour in a home-built chamber which ensured constant humidity. For traditional sample deposition methods, 10 μl of sample was deposited to create a thin protein film covering the prism. The sample was allowed to adsorb for 5 min, then subsequently blotted and rinsed with 5 μl MilliQ water. New background spectra were acquired before each measurement. All spectra were normalised and analysed using OriginPro (Origin Labs). To determine the secondary structure composition of proteins, a second derivative analysis was performed. Spectra were first smoothed by applying a Savitzky-Golay filter. For HDX experiments, the samples were prepared using microfluidic spray deposition. The prism, with deposited protein thin film, was first flushed with $H_2O$ vapour to serve as a baseline, as sample film swelling can affect acquired spectra. Vapour was produced by bubbling $N_2$ gas through $H_2O$ to a home-built chamber around the prism. The input lines to the chamber were then switched to flush the prism with $D_2O$ vapour and measurement started immediately. The integral of the amide II' peak intensity (1450 $cm^{-1}$) was used to monitor exchange over time. The presence of residual $H_2O$ in the chamber was monitored via the peak at ~3300 $cm^{-1}$ to remove the confounding variable of differing $D_2O$ amounts.

## Reporting summary

Further information on research design is available in the Nature Portfolio Reporting Summary linked to this article.

## Data availability

All data is provided in the main text or supplementary data files.

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

## Acknowledgements

We gratefully acknowledge Dr. Heather Greer for assistance with TEM, supported by EPSRC Underpinning Multi-User Equipment Call (EP/P030467/1). We also thank Dr. Michael Metrick for helpful advice regarding FTIR hydrogen-deuterium exchange experiments, and Dr. Selen Manioglu and Prof. Daniel Müller for helpful advice regarding multi-parametric nano-mechanical mapping experiments.

## Author contributions

A.M., F.S.R., and M.V. conceived the project. A.M. designed and performed experiments. Z.T. also performed experiments. S.Q. and P.SGH. provided key reagents. A.M. analysed the data. F.S.R., T.P.J.K., and M.V. provided supervision. A.M., F.S.R., and M.V. wrote the manuscript. All authors discussed the data, edited and approved the final manuscript.

## Competing interests

The authors declare no competing interests.
