## [Transparent Peer Review file · Communications Chemistry]

Nanoscale profiling of evolving intermolecular interactions in ageing FUS condensates

Corresponding Author: Professor Michele Vendruscolo

Version 0:

This manuscript has been previously reviewed at another journal. This document only contains information relating to versions considered at Communications Chemistry.

Reviewer comments:

Reviewer #1

(Remarks to the Author)

We believe that the authors have adequately addressed the concerns raised by me and Reviewers 3 and 4, and recommend the manuscript for acceptance.

They have removed the previous interpretation of the FTIR data suggesting that water remains in the dense phase even after the evaporation of bulk water. Since the signals from NH groups and water overlap, the FTIR region around 3400 cm^{-1} cannot reliably support such a conclusion. Instead, the authors have focused their interpretation on the amide region of the spectrum. Additionally, the helical structure of the protein is further supported by the FTIR spectrum obtained in D_2O .

To the reviewer 4's comment that "If there was no way to perform AFM measurements under solution conditions, I guess the argument could have been made that these experimental conditions were the only accessible ones for this study. But this is not the case. Solution AFM is, while less common, an established technique." The authors have replied that they have clarified that AFM can be performed in solution in line 258. This sentence already appears in line 269 of the previous version. The reviewers are not questioning the use of AFM in the solution state but rather asking what is novel about the method employed, given that solution AFM is an established technique. The authors are therefore expected to clarify what new insights are provided by combining microfluidic spray deposition with solution-state AFM.

However, this concern has already been addressed in the revised manuscript in line 512, where the authors write: "In this work, we presented a characterization of protein condensates on surfaces through a combination of AFM-based nanomechanical mapping and bulk FTIR using a microfluidic spray deposition method. We showed that it is possible to preserve relevant conformational features of FUS condensates on surfaces, thus enabling us to perform high-resolution structural and mechanical characterization of the ageing behavior of these condensates."

This statement demonstrates that the revised manuscript effectively conveys how the combination of AFM and FTIR, when applied to protein droplets prepared via microfluidic spray deposition, allows for structural characterization under conditions that preserve the native conformational features of the proteins.

Overall, all the comments have been addressed, and the manuscript can be accepted for publication.

Reviewer #2

(Remarks to the Author)

The authors have addressed my previous concerns and comments, and I also believe they have adequately addressed the concerns raised by Reviewers 3 and 4. I recommend the manuscript for acceptance.
